# Clinical outcomes of immunomodulatory therapies in the management of COVID-19: A tertiary-care experience from Pakistan

**Noreen Nasir[1], Salma Tajuddin[1], Sarah Khaskheli[2], Naveera Khan[2], Hammad Niamatullah[2], Nosheen Nasir[1]\***

**1** Department of Medicine, Aga Khan University, Karachi, Pakistan, **2** Medical College, Aga Khan University, Karachi, Pakistan

\* Nosheen.nasir@aku.edu

## Abstract

The pharmacological management of COVID-19 has evolved significantly and various immunomodulatory agents have been repurposed. However, the clinical efficacy has been variable and a search for cure for COVID-19 continues. A retrospective cohort study was conducted on 916 patients hospitalized with polymerase chain reaction (PCR)-confirmed COVID-19 between February 2020 and October 2020 at a tertiary care academic medical center in Karachi, Pakistan. The median age was 57 years (interquartile range (IQR) 46–66 years). The most common medications administered were Methylprednisolone (65.83%), Azithromycin (50.66%), and Dexamethasone (46.6%). Majority of the patients (70%) had at least two or more medications used in combination and the most frequent combination was methylprednisolone with azithromycin. Overall in-hospital mortality was 13.65% of patients. Mortality was found to be independently associated with age greater than or equal to 60 years (OR = 4.98; 95%CI: 2.78–8.91), critical illness on admission (OR = 13.75; 95%CI: 7.27–25.99), use of hydrocortisone (OR = 12.56; 95%CI: 6.93–22.7), Ferritin> = 1500(OR = 2.07; 95%CI: 1.18–3.62), Creatinine(OR = 2.33; 95%CI: 1.31–4.14) and D-Dimer> = 1.5 (OR = 2.27; 95%CI: 1.26–4.07). None of the medications whether used as monotherapy or in combination were found to have a mortality benefit. Our study highlights the desperate need for an effective drug for the management of critical COVID-19 which necessitates usage of multiple drug combinations in patients particularly Azithromycin which has long term implications for antibiotic resistance particularly in low-middle income countries.

## Introduction

As of August 2021; Corona Virus Disease 2019 (COVID-19) has caused more than 4 million deaths worldwide [1]. The treatment strategy for COVID-19 involves providing general supportive care, respiratory support, symptomatic treatment, nutritional support, and psychological intervention [2]. Various pharmacological agents have reduced disease severity and mortality [3].

**Data Availability Statement:** Data files are deposited to a stable, public repository. DOI is provided as follows: Nasir, Nosheen (2021),

"Deidentified data", Mendeley Data, V1, doi: 10.17632/wh44wh544p.1.

**Funding:** The author(s) received no specific funding for this work.

**Competing interests:** The authors have declared that no competing interests exist.

Numerous observational studies and clinical trials have evaluated the effectiveness of several drug classes, including antivirals, anti-inflammatory agents, immunomodulators, corticosteroids, novel therapeutic agents, and novel small molecule inhibitors or Janus kinase (JAK) inhibitors [4,5]. Therapies targeting viral replication, such as remdesivir, have shown to be effective in shortening the time to recovery in patients suffering from COVID-19 [6]. The anti-interleukin-6 (IL-6) receptor antibody tocilizumab reportedly reduced the likelihood of mechanical ventilation and death [7–9]. The use of glucocorticoids in hospitalized patients with COVID-19 has been associated with improved clinical outcomes and decreased mortality [10,11]. In addition, the efficacy of other pharmacological agents used in the treatment of COVID-19 has been studied notably azithromycin [12,13] and hydroxychloroquine [14,15]. However, the list is exhaustive as the search for cure continues. Of all the experimental agents used for COVID 19, the most substantial evidence has emerged for IL-6 inhibition with Tocilizumab and Sarilumab [16]. Both drugs have World health organization (WHO) endorsement for management of Critical COVID as well as food and drug administration (FDA) approval for Tocilizumab [17].

The pharmacological management of COVID-19 has evolved significantly, leading to novel therapeutics, including monoclonal antibodies and Janus-kinase inhibitors. However, low- and middle-income countries (LMICs) like Pakistan face significant challenges in managing COVID-19, given scarce drug procurement resources for inpatient care [18]. While considerable evidence has evolved in favor of, and various immunomodulatory agents have been repurposed, evidence on their clinical efficacy remains to be established [19]. A concerted effort is needed to evaluate the potential of these agents. Therefore, we aim to describe the patterns of pharmacotherapy for COVID-19, emphasizing the number of drug combinations used in a given time in patients with severe COVID-19.

## Materials and methods

We conducted a retrospective cohort study in COVID-19 patients admitted from February 2020 to October 2020. Ethical approval was obtained before the commencement of the study from the Aga Khan University ethics review committee (AKU ERC Reference number: 2020-4939-11055). All adult patients greater than or equal to 18 years of age who had tested positive for SARS-CoV-2 by nasopharyngeal reverse transcriptase polymerase chain reaction (RT-PCR) were included. Cytokine storm (CS) or hyperinflammation was defined as serum C-reactive protein (CRP) $\geq$100 mg/L, ferritin $\geq$900 ng/mL, or both. Non-severe COVID-19 was described as having symptoms of fever, cough, or other symptoms with radiographic evidence of pneumonia but no hypoxia or evidence of CS. Severe disease was defined as having respiratory distress, respiratory rate >30 breaths/min on rest, Oxygen saturations of <93%, and Partial Pressure of Oxygen to Fraction of Inspired Oxygen Ration (PaO2/Fio2) <300mmHg. The acute disease was defined as respiratory failure requiring mechanical ventilation and shock with or without multiorgan dysfunction requiring Intensive Care Unit (ICU) monitoring. Data was collected on a structured proforma from the hospital information management system (HIMS) Department at Aga Khan University hospital (AKUH) on patients admitted with COVID-19. Outcome variables included in-hospital mortality and length of stay.

### Statistical analysis

Median with interquartile range (IQR) was computed for age and length of hospital stay (LOS), and frequencies (percentage) were calculated for variables such as gender and outcomes. Continuous variables were assessed after appropriate transformation into categorical

variables depending on clinical relevance. χ2 test of independence was performed for comparison of results. Univariable and multivariable logistic regression analysis was performed to determine risk factors associated with in-hospital mortality. Adjusted odds ratios with Confidence Intervals were reported. We included interaction terms to identify any interaction between different therapeutic options. STATA Version 12.1 was used for data analysis.

Furthermore, the dataset included all date-wise records for all drugs prescribed and lab tests conducted on all hospitalized COVID-19 patients at AKUH between February 2020 and October 2020. Pattern recognition was used to get the primary filtered name for each drug. The filtered primary terms were assigned to one or none of the set categories: Hydrocortisone; Remdesivir; Hydroxychloroquine; Tocilizumab; Dexamethasone; Azithromycin; and Methylprednisolone. The selected combination of drugs was then presented using an upset diagram. Time-based graphs were created to study the impact of medications on selected labs: Ferritin; C—reactive protein; lactate dehydrogenase (LDH) and D-Dimer. To make the chart of each drug, relative lab results were calculated, keeping the first day of drug prescription as the reference day. Results 7-day before and 7-days after each drug administration were considered. The patients were pooled into two groups based on COVID-19 severity, into severe and non-severe patients. Severe patients were identified based on either Ferritin lab results greater than 1500 or C-reactive protein greater than 150 anytime during their stay. Stratified line graphs were created using the median results of each day for each patient group. To mitigate the impact of the non-daily test, a geom_smooth function was used on the line graphs. All analysis was done in R and Stata ver 12.1.

## Results

### Patient demographics

We collected data from 916 patients hospitalized with PCR-confirmed COVID-19 between February 2020 and October 2020. The median age of the included patients was 57 years (IQR 46–66 years). Most patients (68.88%) were over the age of 50 years and majority were male (66.48%). Co-morbid data was available for n = 700 patients. The most frequent co-morbids were Diabetes mellitus (DM) in 44.5%, hypertension in 44% and ischemic heart disease (IHD) in 12.2%. The most common medications administered were Methylprednisolone (65.83%), Azithromycin (50.66%), and Dexamethasone (46.6%) (Fig 1). In-hospital mortality was seen in 13.65% of patients, whereas 78.82% were alive at discharge, and 7.53% left the hospital against medical advice.

A decline in median inflammatory parameters between Day 1 and Day 3 of admission and treatment initiation was observed for C-reactive protein (71.92 mg/L (IQR 29.85–150.31) to 26.87 mg/L (IQR 12.94–54.76)), Procalcitonin (0.16 ng/ml (IQR 0.08–0.50) to 0.12 ng/ml (IQR 0.06–0.39)) and percentage of lymphocytes (11.3(IQR 6.6–18.6) to 9.8 (IQR 6.1–15.1)). As opposed to this; a rise in median values of D-dimer(1.1 mcg/mL (IQR 0.6–2.9) to 1.2 mcg/mL (IQR 0.6–3.3), ferritin (787.45 ng/ml(IQR 379.85–1368.25) to 909.10 ng/ml (IQR 524.90–1382.90), LDH (435 I.U/L (IQR 336–583) to 438 I.U/L (IQR 340–572) and Creatinine (0.9 mg/dL (IQR 0.7–1.2) to 1.0 mg/dL (IQR 0.7–1.4) was observed between Day 1 and Day 3 of admission despite initiation of medication and this difference was more marked for those with severe COVID-19 (Fig 2).

### Logistic regression analyses

On univariate analysis; a CRP $> = 100$ ($P<0.001$), Ferritin$> = 1500$ ($P<0.001$), D-Dimer $>1.5$ ($P<0.001$) and LDH$> = 250$ ($P = 0.023$) held a statistically significant association with death (Table 1). Among co-morbid conditions, ischemic heart disease was significantly associated

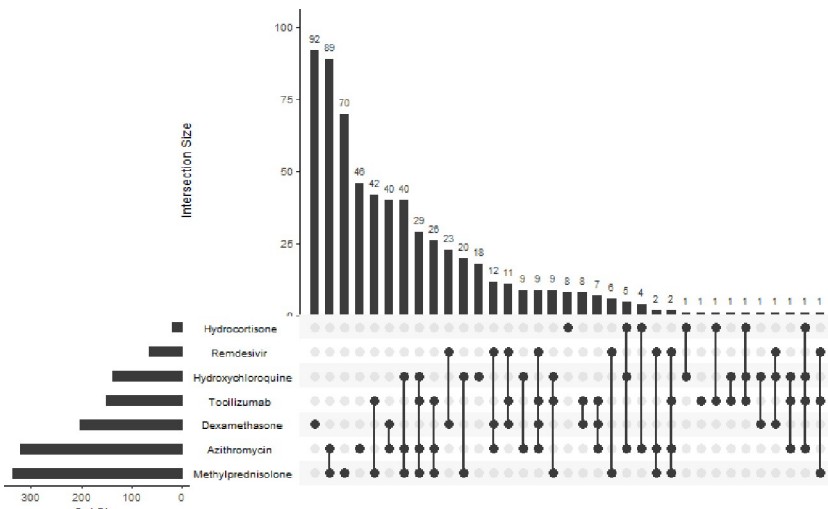

**Fig 1. Combinations of drugs used in management of COVID-19.** In this Upset graph, each row represents a drug. Each column represents a combination of drugs. A filled dot indicates that drug is included in the column combination.

with mortality ($P = 0.038$). Among medications, administration of Hydrocortisone ($P<0.001$), Hydroxychloroquine ($P = 0.013$), Methylprednisolone ($P = 0.008$) and Tocilizumab ($P = 0.021$) were associated with ICU admission and death. Moreover, at least 4 or more drugs were being used in combination in patients who died ($<0.001$). However in multivariate logistic regression analysis, mortality was found to be independently associated with age greater than or equal to 60 years (aOR = 4.98; 95%CI: 2.78–8.91), critical illness on admission (aOR = 13.75; 95%CI: 7.27–25.99), use of hydrocortisone (aOR = 12.56; 95%CI: 6.93–22.7), Ferritin$> = 1500$(aOR = 2.07; 95%CI: 1.18–3.62), Creatinine(aOR = 2.33; 95%CI: 1.31–4.14) and D-Dimer$> = 1.5$ (aOR = 2.27; 95%CI: 1.26–4.07) (Table 2). None of the medications whether used as monotherapy or in combination were found to have a mortality benefit and none of the co-morbids were found to be independently associated with death.

## Discussion

The greatest concern in management of critical COVID-19 patients is survival. Our study sought to investigate the patterns in pharmacotherapy and the factors associated with in-hospital mortality. Notable findings included an increased risk of death with advanced age, critical illness and acute kidney injury based on serum creatinine greater than 1.5 mg/dl as well as biomarkers such as Ferritin and D-Dimer. Moreover, we found a trend towards greater number of medications used in combination in critically ill patients.

Scientific evidence on pharmacological management of COVID-19 continues to evolve and is in a state of flux [20]. Following the pandemic, hydroxychloroquine utilization increased rapidly due to its immunomodulatory properties, but it declined sharply after May 2020 due to safety concerns and a lack of efficacy data [21]. In contrast, the use of dexamethasone and corticosteroids progressively increased during 2020 based on findings from the randomized evaluation of COVID-19 therapy (RECOVERY Trial) [10,22,23]. Corticosteroid therapy is used in COVID 19 to mitigate the inflammatory response in the lungs [24]. Methylprednisolone was the most commonly administered medication in COVID-19 patients in our study cohort although it did not improve survival. A randomized controlled trial (RCT) involving severe hospitalized COVID-19 patients receiving Methylprednisolone versus standard of care showed

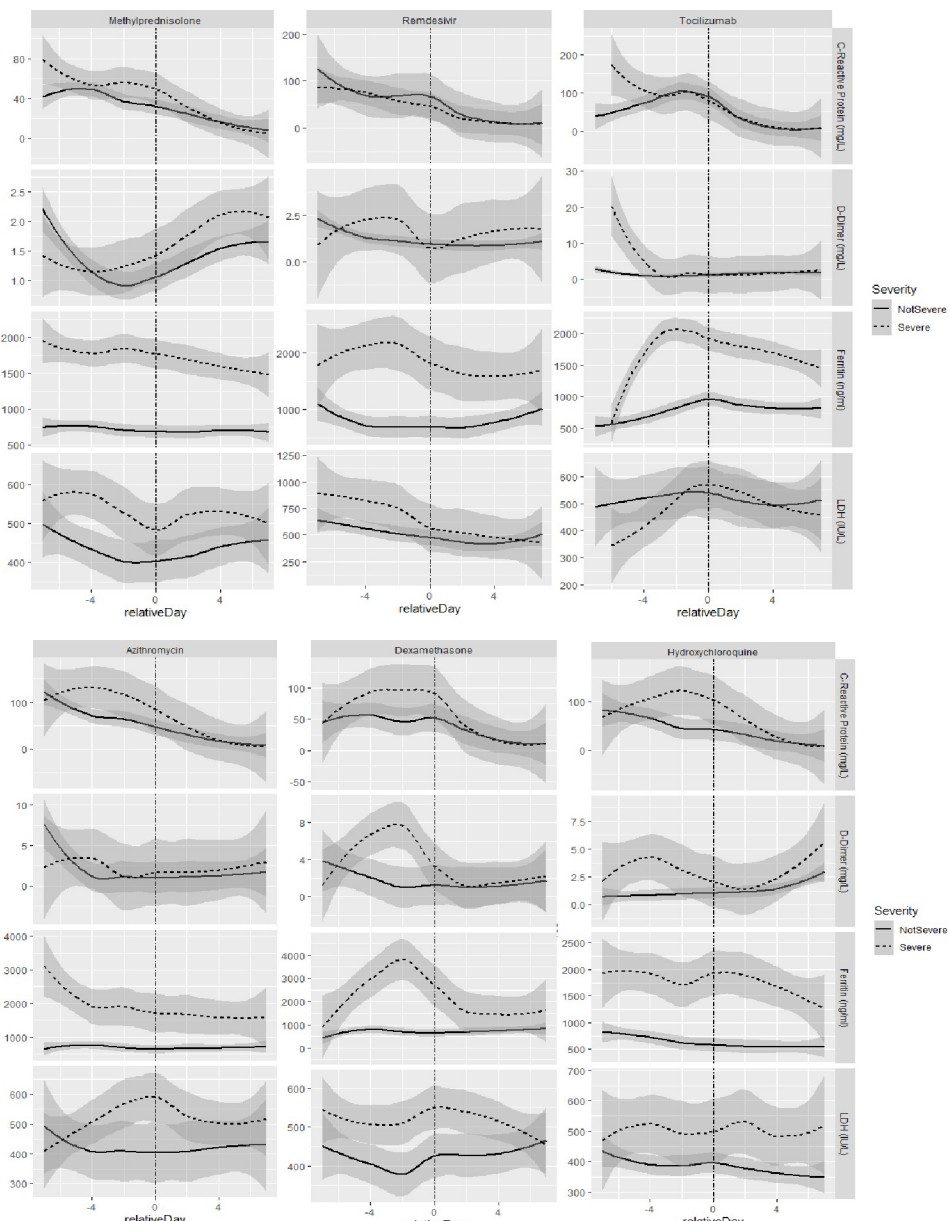

**Fig 2. Median laboratory values of inflmmatory markers in relation to administration of drug stratified by severity of COVID-19.** The graphs show the median lab values of patient in Severe and Non-severe categories against days relative to the day of the administration of the drug. Each graph has been smoothed using a regression function to mitigate for sparingly conducted lab tests.

that patients in the methylprednisolone group had significantly increased survival times and higher rates of clinical improvement compared with patients in the standard care group [25]. IL-6 inhibition with Tocilizumab and Sarilumab has changed the treatment paradigm for severe and critically ill patients with COVID 19. An insight into the pathophysiology of cytokine storms led to a corresponding biochemical evaluation of IL-6, LDH, Ferritin, C reactive protein, D-Dimer in susceptible patients [26]. Most of the patients who were managed in special and intensive care units had significant elevations of cytokine release syndrome (CRS) biomarkers. These patients received Tocilizumab as salvage therapy, although it turned out that

**Table 1. Comparison of COVID-19 patients who died and who survived.**

| Variables | Died (n = 125) | Alive (n = 722) | p-value |
|---|---|---|---|
| Median Age (IQR) years | 65 (56–75) | 55 (44–65) | <0.001 |
| Age Range n (%) | | | <0.001 |
| 18–29 | 3 (2.4) | 35 (4.9) | |
| 30–49 | 10 (8.0) | 219 (30.3) | |
| 50–69 | 60 (48.0) | 357 (49.5) | |
| > = 70 | 52 (41.6) | 111 (15.4) | |
| Gender n (%) | | | 0.342 |
| Male | 87 (69.6) | 471 (65.2) | |
| Female | 31 (30.4) | 251 (34.7) | |
| Critical illness | 68 (58.6) | 57 (7.12) | <0.001 |
| Laboratory parameters | | | |
| CRP <100 (Ref) | 54 (43.2) | 438 (60.7) | <0.001 |
| CRP > = 100 | 71 (56.8) | 284 (39.3) | |
| Ferritin <1500 (Ref) | 72 (57.6) | 546 (75.6) | <0.001 |
| Ferritin> = 100 | 53 (42.4) | 176 (24.4) | |
| D-Dimer <1.5 (Ref) | 30 (24.0) | 412 (57.1) | <0.001 |
| D-Dimer >1.5 | 95 (76.0) | 310 (42.9) | |
| LDH <250 (Ref) | 3 (2.4) | 57 (7.9) | 0.023 |
| LDH> = 250 | 122 (97.6) | 665 (92.1) | |
| Medications n (%) | | | |
| Azithromycin | 60 (48.0) | 363 (50.2) | 0.638 |
| Dexamethasone | 53 (42.4) | 337 (46.7) | 0.376 |
| Hydrocortisone | 73 (58.4) | 38 (5.3) | <0.001 |
| Hydroxychloroquine | 33 (26.4) | 123 (17.0) | 0.013 |
| Methylprednisolone | 96 (76.8) | 467 (64.6) | 0.008 |
| Remdesivir | 17 (13.6) | 91 (12.6) | 0.758 |
| Tocilizumab | 26 (20.8) | 94 (13.0) | 0.021 |
| Number of drug combinations n (%) | | | <0.001 |
| <4 (Ref) | 88 (11.9) | 653 (88.1) | |
| > = 4 | 37 (34.9) | 69 (65.1) | |

Abbreviations: IQR: Interquartile range; CRP: C-reactive protein; LDH: Lactate dehydrogenase.

many patients still had poor outcomes due to the severity of the disease. Moreover, raised D-dimer level, Ferritin, and C-reactive protein (CRP) levels were also associated with mortality. In a case-control study, D-dimer levels correlated with disease severity and were a reliable prognostic marker for in-hospital mortality in patients admitted for COVID-19 [27]. In another study, the hospitalization-wide median CRP was significantly higher amongst the patients who died than those who survived [28].

The in-hospital mortality for our patient population was 13.65%, which was similar to the global trends of in-hospital mortality among hospitalized COVID-19 patients [29]. Multivariate logistic regression results showed that age and critical illness in our patient cohort were independently associated with mortality. This finding is similar to a retrospective cohort study of hospitalized COVID-19 patients treated in 592 hospitals in the United States which found that in-hospital mortality rates were more significant in older patients [30]. A review by Ejaz et al. describes that patients with co-morbid conditions were at higher risk of severe illness and adverse outcomes [31]. In our study we did not find co-morbid conditons such as DM,

**Table 2. Multivariable logistic regression model for factors associated with in-hospital mortality.**

| Variable | Categories | OR | 95% CI | p-value |
|---|---|---|---|---|
| Age | < 60 years (Ref) | 1 | | |
| | > = 60 years | 4.98 | 2.78–8.91 | <0.001 |
| Critical illness | Absent (Ref) | 1 | | |
| | Present | 13.75 | 7.27–25.99 | < 0.001 |
| Use of Hydrocortisone | Absent (Ref) | 1 | | |
| | Present | 12.56 | 6.93–22.7 | <0.001 |
| Creatinine | < 1.5 (Ref) | 1 | | |
| | > = 1.5 | 2.33 | 1.31–4.14 | 0.004 |
| Ferritin | < 1500(Ref) | 1 | | |
| | > = 1500 | 2.07 | 1.18–3.62 | 0.011 |
| D-Dimer | < 1.5 (Ref) | 1 | | |
| | > = 1.5 | 2.27 | 1.26–4.07 | 0.006 |

hypertension and IHD to be associated with poor prognosis although there was a trend towards higher mortality in IHD patients in univariate analysis. This is similar to large study from Bergamo, Italy by Novelli et al. in which co-morbids were not found to be significantly associated with mortality [32].

Patterns of pharmacotherapy, specifically the drug regimens and combinations received by hospitalized patients during their stay, were studied by presenting a selected combination of drugs using an upset diagram. The most frequently used drug combinations were Methylprednisolone and Azithromycin followed by methylprednisolone and Tocilizumab. Fewer studies have shown data on use of combination treatment in COVID-19. In one retrospective cohort study, the top three treatment combinations were 'Anticoagulation Only,' 'Anticoagulation and Hydroxychloroquine' and 'Anticoagulation and Corticosteroids' [33].

More than half of the patients in our cohort who died developed critical illness (58.6%). The patients who survived received < four-drug combinations. On the other hand, the majority of patients who died received > = four-drug regimen. These trends are consistent with the findings from a study on pharmacological treatment patterns by COVID-19 severity. It showed that patients with greater disease severity (intubated, requiring mechanical ventilation) were 3.53 times more likely to receive a medication to treat COVID-19 [34].

Thus, we report one of the most extensive studies on the inpatient clinical management of COVID-19 from Pakistan, a lower-middle-income country in South Asia. We hope that it will inform best practices for other developing countries and drive a search for more potent and effective pharmacotherapeutics, and for the scientific community to include patients from developing countries in their ongoing clinical trials. The main limitations of this study are that our findings are based on results from a single health care facility and we have used retrospective design which carries risk of missing data. Therefore, larger multi-center studies are required to validate and strengthen these findings further.

## Conclusion

Our study highlights the desperate need for an effective drug for the management of critical COVID-19 which necessitates usage of multiple drug combinations in patients particularly Azithromycin which has long term implications for antibiotic resistance particularly in low-middle income countries.

## Acknowledgments

We acknowledge the contribution of Health Information Management system and Pharmacy department at Aga Khan University in providing data for this study.

## Author Contributions

**Conceptualization:** Noreen Nasir, Nosheen Nasir.

**Data curation:** Salma Tajuddin, Naveera Khan, Hammad Niamatullah, Nosheen Nasir.

**Formal analysis:** Salma Tajuddin, Nosheen Nasir.

**Investigation:** Noreen Nasir, Salma Tajuddin, Sarah Khaskheli, Hammad Niamatullah.

**Methodology:** Noreen Nasir, Salma Tajuddin, Sarah Khaskheli, Naveera Khan, Hammad Niamatullah, Nosheen Nasir.

**Project administration:** Nosheen Nasir.

**Supervision:** Noreen Nasir.

**Writing – original draft:** Noreen Nasir, Sarah Khaskheli, Naveera Khan, Nosheen Nasir.

**Writing – review & editing:** Noreen Nasir, Salma Tajuddin, Sarah Khaskheli, Naveera Khan, Hammad Niamatullah, Nosheen Nasir.

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
