## [Decision Letter · Decision Letter 0]

14 Dec 2021

PONE-D-21-26285Clinical outcomes of immunomodulatory therapies in the management of COVID-19: a tertiary-care experience from PakistanPLOS ONE

Dear Dr. Nosheen Nasir

Thank you for submitting your manuscript to PLOS ONE. After careful consideration, we feel that it has merit but does not fully meet PLOS ONE’s publication criteria as it currently stands. Therefore, we invite you to submit a revised version of the manuscript that addresses the points raised during the review process.

We look forward to receiving your revised manuscript.

Kind regards,

Eleni Magira

Academic Editor

PLOS ONE

Journal Requirements:

2. Please ensure that you have specified (1) whether consent was informed, (2) what type you obtained (for instance, written or verbal, and if verbal, how it was documented and witnessed). If your study included minors, state whether you obtained consent from parents or guardians. If the need for consent was waived by the ethics committee and (3) If you are reporting a retrospective study of medical records or archived samples, please ensure that you have discussed whether all data were fully anonymized before you accessed them and/or whether the IRB or ethics committee waived the requirement for informed consent. If patients provided informed written consent to have data from their medical records used in research, please include this information.

5. We note you have included a table to which you do not refer in the text of your manuscript. Please ensure that you refer to Table 1 in your text; if accepted, production will need this reference to link the reader to the Table

**Comments to the Author**

1. Is the manuscript technically sound, and do the data support the conclusions?

Reviewer #1: Yes

2. Has the statistical analysis been performed appropriately and rigorously? 

Reviewer #1: Yes

3. Have the authors made all data underlying the findings in their manuscript fully available?

Reviewer #1: Yes

4. Is the manuscript presented in an intelligible fashion and written in standard English?

Reviewer #1: Yes

5. Review Comments to the Author

Reviewer #1: In this tertiary-care experience from Pakistan, the authors reported clinical outcomes of immunomodulatory therapies in the management of COVID-19.

All acronyms should be explained, to help the readers to understand. Please verify all acronyms.

The section conclusion should mainly report the results of the study. On the contrary, in this version it repeats the Introduction, for this reason it should be shortened.

Due to the rapid changes in this field I suggest to update some reference:

-for example the number 3 (year 2020) should be replaced with the recent paper by Patrucco et al. Pol Arch Intern Med. 2021 Sep 30;131(9):854-861.

A relevant data, not presented in this study, is the presence of comorbidities of these patients.

This is very important because several studies have shown that comorbidities did influence the prognosis (Ejaz et al. J Infect Public Health. 2020;13(12):1833-1839) while other did not (for example Novelli et al. Panminerva Med. 2021;63(1):51-61). If the authors do not have these data, they should report it as study limitation and specify the above reported concept with references.

A limitation that the authors hould be added regards the restrospective design of the study with the risk of missing data.

6. PLOS authors have the option to publish the peer review history of their article (what does this mean?). If published, this will include your full peer review and any attached files.

Reviewer #1: **Yes: **Rinaldo Pellicano

---

## [Author Response · Author response to Decision Letter 0]

24 Dec 2021

RESPONSE TO REVIEWER COMMENTS

Reviewer: 1

Comments to the Author

Comment #1: In this tertiary-care experience from Pakistan, the authors reported clinical outcomes of immunomodulatory therapies in the management of COVID-19.

All acronyms should be explained, to help the readers to understand. Please verify all acronyms.

Response 1: Thanks for the comment. We have edited to include and verify all acronym descriptions.

Comment #2:

The section conclusion should mainly report the results of the study. On the contrary, in this version it repeats the Introduction, for this reason it should be shortened.

Response #2: We have revised conclusion section as per recommendation of reviewer

Comment #3

Due to the rapid changes in this field I suggest to update some reference:

-for example the number 3 (year 2020) should be replaced with the recent paper by Patrucco et al. Pol Arch Intern Med. 2021 Sep 30;131(9):854-861.

Response #3: We have updated reference as above

Comment #4

A relevant data, not presented in this study, is the presence of comorbidities of these patients.

This is very important because several studies have shown that comorbidities did influence the prognosis (Ejaz et al. J Infect Public Health. 2020;13(12):1833-1839) while other did not (for example Novelli et al. Panminerva Med. 2021;63(1):51-61). If the authors do not have these data, they should report it as study limitation and specify the above reported concept with references.

Response #4:We have added our comorbids data and added findings to results section. We have added the above references with respect to our findings.

Comment #5

A limitation that the authors hould be added regards the restrospective design of the study with the risk of missing data.

Response #5

We have added the above to limitation as recommended.

---

## [Decision Letter · Decision Letter 1]

31 Dec 2021

Clinical outcomes of immunomodulatory therapies in the management of COVID-19: a tertiary-care experience from Pakistan

PONE-D-21-26285R1

Dear Dr. Nosheen Nasir

We’re pleased to inform you that your manuscript has been judged scientifically suitable for publication and will be formally accepted for publication once it meets all outstanding technical requirements.

Kind regards,

Eleni Magira

Academic Editor

PLOS ONE

Additional Editor Comments (optional):

Reviewers' comments:

Reviewer's Responses to Questions

**Comments to the Author**

1. If the authors have adequately addressed your comments raised in a previous round of review and you feel that this manuscript is now acceptable for publication, you may indicate that here to bypass the “Comments to the Author” section, enter your conflict of interest statement in the “Confidential to Editor” section, and submit your "Accept" recommendation.

Reviewer #1: All comments have been addressed

2. Is the manuscript technically sound, and do the data support the conclusions?

Reviewer #1: Yes

3. Has the statistical analysis been performed appropriately and rigorously? 

Reviewer #1: Yes

4. Have the authors made all data underlying the findings in their manuscript fully available?

Reviewer #1: Yes

5. Is the manuscript presented in an intelligible fashion and written in standard English?

Reviewer #1: Yes

6. Review Comments to the Author

Reviewer #1: I have read the new version of this manuscript. The authors modified the text according to the requests. Hence, I do not have further questions

7. PLOS authors have the option to publish the peer review history of their article (what does this mean?). If published, this will include your full peer review and any attached files.

---

## [Editor Report · Acceptance letter]

20 Jan 2022

PONE-D-21-26285R1 

Clinical outcomes of immunomodulatory therapies in the management of COVID-19: a tertiary-care experience from Pakistan 

Dear Dr. Nasir:

I'm pleased to inform you that your manuscript has been deemed suitable for publication in PLOS ONE. Congratulations! Your manuscript is now with our production department. 

Kind regards, 

on behalf of

Dr. Eleni Magira 

Academic Editor

PLOS ONE